# Predatory Dogs as Drivers of Social Behavior Changes in the Central Himalayan Langur (*Semnopithecus schistaceus*) in Agro-Forest Landscapes

**DOI:** 10.3390/biology13060410

**Published:** 2024-06-04

**Authors:** Himani Nautiyal, Virendra Mathur, Kimaya Hemant Gajare, Julie Teichroeb, Dipto Sarkar, Rui Diogo

**Affiliations:** 1College of Medicine, Howard University, 520 W St, NW, Washington, DC 20059, USA; rui.diogo@howard.edu; 2Department of Anthropology, University of Toronto, Scarborough 1265 Military Trail, Toronto, ON M1C 1A4, Canada; viren.mathur@mail.utoronto.ca (V.M.); julie.teichroeb@utoronto.ca (J.T.); 3Bharatiya Vidya Bhavan’s, Bhavan’s College, University of Mumbai, Andheri (w), Mumbai 400 058, India; kimayagajare@gmail.com; 4Department of Geography and Environmental Studies, Carleton University, 1125 Colonel By Drive, Ottawa, ON K1S 5B6, Canada; dipto.sarkar@carleton.ca

**Keywords:** human-induced rapid environmental change, human–wildlife interactions, landscape of fear, predator–prey systems, resource availability

## Abstract

**Simple Summary:**

In human-modified landscapes, wildlife species are increasingly pushed into unfamiliar territories due to habitat fragmentation caused by urbanization and agricultural development. This shift poses various challenges for these species, including interactions with humans and predators like free-ranging dogs. To understand how wildlife adapts to such circumstances, we examined the behavioral strategies of the Central Himalayan langur (CHL) in response to the presence of predatory dogs and humans. Over two years, we observed a group of CHLs living in a fragmented landscape, frequently foraging on agricultural and fodder crops, and thus encountering villagers and free-ranging dogs. Our research found significant changes in the CHLs’ major activities, such as resting, feeding, moving, and socializing in the presence of dogs and humans. Specifically, CHLs performed increased movement and feeding and reduced social activity. These behavioral adaptations likely aim to optimize survival in a challenging habitat. However, such modifications in the long term could lead to issues for both humans and wildlife. Increased movement and feeding in agricultural areas may lead to high negative interactions with farmers and dogs, while reduced social activity could impact social dynamics and reproductive success. Implementing sustainable agricultural practices, providing alternative livelihood options, promoting coexistence strategies, and engaging local communities in conservation efforts will be essential steps toward minimizing these concerns.

**Abstract:**

Globally, habitat fragmentation has increased the proximity between wildlife, humans, and emerging predators such as free-ranging dogs. In these fragmented landscapes, encounters between primates and dogs are escalating, with primates often falling victim to dog attacks while navigating patchy landscapes and fragmented forests. We aim to investigate how these primates deal with the simultaneous threats posed by humans and predators, specifically focusing on the adaptive strategies of Central Himalayan langur (CHL) in the landscape of fear. To address this, we conducted a behavioral study on the CHL in an agro-forest landscape, studying them for a total of 3912 h over two consecutive years. Our results indicate that, compared to their most common resting behavior, CHLs allocate more time to feeding and locomotion, and less time to socializing in the presence of humans and predatory dogs. Additionally, they exhibit increased feeding and locomotion and reduced social behavior in agro-forest or open habitats. These behavioral patterns reflect adaptive responses to the landscape of fear, where the presence of predators significantly influences their behavior and resource utilization. This study suggests measures to promote coexistence between humans and wildlife through the integration of effective management strategies that incorporate both ecological and social dimensions of human–wildlife interactions.

## 1. Introduction

Domestic dogs *Canis familiaris* are globally the most widespread human-introduced carnivores [1]. Originating roughly 14,000 years ago [2], the dog population grew alongside the human population, with estimates nearing a billion worldwide [3]. Through their long-term association with humans, dogs have become integral members of human society, fulfilling various roles, such as pets, protectors, and guards for agriculture and livestock [4,5]. Across the globe, dogs exhibit varying degrees of free-ranging behavior, bringing them into interactions with wildlife on multiple levels [1]. Particularly in developing nations, the threat of free-ranging dogs escalates as human populations expand into natural areas in search of suitable agricultural land and natural resources [6]. Despite their long history as human companions and workers, the impact of dogs on wildlife hinges on their management practices, whether they are fully domesticated and leashed, roam freely, rely on human-provided food, or lead independent lives. Additionally, the proximity of dogs to natural reserves or urban pockets of wildlife plays a significant role in the frequency and nature of interactions with wildlife [6,7].

Despite their morphological differences from wolves due to domestication, dogs retain characteristics of the carnivore guild [1,4]. In certain regions, dogs have emerged as the most abundant carnivores, disrupting ecosystems [8] and negatively impacting wildlife. Dog behaviors, including direct predation, harassment, and competition, can lead to fear-induced behavioral changes in wildlife and affect their resource base and ultimately their fitness. In addition, the mere presence of dogs can lead to disease transmission [9,10,11]. Prominent review articles provide insightful analyses of the complex dynamics of human–dog–wildlife interactions, highlighting the need for effective management strategies to mitigate adverse effects on biodiversity [1,4,7,12,13]. However, there remains a need to understand how dogs are increasingly impacting individual species of wildlife as humans expand their agricultural areas.

Primate species represent one of the most diverse groups of mammals, totaling 504 species [14]. Unfortunately, approximately 65% of these species are currently under threat of extinction due to habitat loss and resource depletion [15]. Primate habitats are becoming significantly fragmented as a result of rising human demand for agricultural land, industrial growth, and the extraction of natural resources (e.g., mining, fuelwood), [16,17]. Over 50% of tropical and subtropical forest habitats, which are home to a large diversity of primates worldwide, have become fragmented [16]. In these fragmented landscapes, encounters between primates and invasive species, notably dogs, are growing, as primates often fall victim to dog attacks while navigating patchy landscapes and fragmented forests [18,19]. For example, dogs have been hunting and eating white-tailed small-eared Galagos *Otolemur garnettii lasiotis* in Kenya’s fragmented landscapes [19]. Similarly, in Madagascar, dogs have also been observed preying on endangered wild ring-tailed lemurs *Lemur catta* [20], and similar incidents involving langurs and macaques have been reported from comparable landscapes in Asia [21,22,23,24]. The presence of predators, including dogs, profoundly influences primate behavior and activity patterns. Primates, like many other species, have adapted to minimize the risk of predation through behavioral strategies such as alarm calls, vigilance, and counter-attacking [25,26]. Studies conducted in various ecosystems have shown how the perceived risk of predation alters vigilance behavior [27], movement patterns [28], foraging strategies [29], and social dynamics [30]. 

Anthropogenic impacts on animal behavior result from both direct human disturbances, like fragmentation, and indirect disturbances, such as the introduction of alien species [31]. The initial reactions of animals to these changes are often behavioral, helping them adapt to new conditions [32,33,34]. Primates living in anthropogenically disturbed areas regularly show changes in habitat use, foraging, and activity budget [35,36]. For instance, common chimpanzees *Pan troglodytes verus* in Guinea, West Africa, have shown a preference for traveling, resting, and socializing in habitat types with lower levels of human-induced pressure [35]. Compared to natural processes, anthropogenic pressures expose animals to environments they have not experienced in their evolutionary history, and behavioral changes are occurring at faster rates [37]. Even though these behavioral adaptations might have short-term benefits, they may also disrupt other long-evolved, fitness-related behaviors (e.g., social structure, reproductive strategies), which could negatively impact species persistence and have cascading effects on ecosystems [38,39,40]. 

Shared agroforest environments offer a complex web of opportunities and costs to wildlife. Our study species, Central Himalayan langurs (CHLs), inhabit Himalayan high-altitude agroforest landscapes. CHLs dwell within fragmented natural oak forests and rely on seasonal crops (May–June and September–October), as well as on the leaves of fodder trees that grow along the edges of agricultural fields year-round [41,42]. However, foraging within agricultural fields exposes langurs to their main predator, feral dogs [18,43], and increases the likelihood of frequent negative encounters with local people [40,43]. Predator presence can create a ‘landscape of fear’ and can influence the behavior and distribution of prey species within an ecosystem. Thus, ecological dynamics for CHLs in such agroforest landscapes are not solely driven by the availability of resources, but also by the fear of predation and the strategies animals employ to mitigate that risk [44,45,46,47]. By understanding how animals adapt their behavior to minimize risks while maximizing opportunities in shared landscapes, we can develop more effective approaches for managing human–wildlife interactions and foster sustainable coexistence.

While a growing number of case studies describe behavioral responses to human-induced rapid environmental change in specific systems [46], our understanding of such changes due to predatory feral dogs remains limited. It is well established that the presence of dogs alters the spatio-temporal behavior of wildlife [47], escalating competition for space and resources [48], which leads to broader negative consequences such as predation, the transmission of diseases to wildlife, the disturbance of natural habitats, and the potential for hybridization with native species [4,7,12,13,17]. However, to our knowledge, the direct influence of these impacts on animal behavior patterns has not been extensively explored in previous studies.

This study explores how the fear landscape, shaped by both human and predatory dog presence, influences the activity patterns of CHLs in agroforest environments. Our goal is to gain insight into the complex structure of fear behavior and identify specific adaptations in response to perceived threats. We hypothesize that (1) the presence of dogs and humans induces a disturbance in the behavior of CHLs, and that (2) habitats associated with these fear landscapes impact CHL behavior patterns. Consequently, we predict that in fear landscapes, CHLs will modify their activity patterns to reduce the likelihood of encounters with dogs and humans. 

## 2. Materials and Methods

### 2.1. Study Site and Population

The study was conducted from May 2022 to December 2023 in Mandal Valley, Chamoli, Uttarakhand, Garhwal Himalayas, India (coordinates: 30°28′04″ N 79°16′31″ E; elevation: 1500–1800 m above sea level). The habituation process for the study group, “S Group”, began in 2014. Since then, this group has remained the subject of continuous studies, up to the present day. Consequently, the study group has become habituated to the presence of researchers. The total home range of the S Group encompasses both natural forests and agricultural fields, frequently bringing them into proximity with humans and predatory domestic dogs (Figure 1). For further details on previous studies conducted on the S Group, please refer to the following studies [18,40,41,42,49,50]. Every member of the group, as well as each newcomer, was individually identified and monitored throughout the duration of the study. During the observation time for this study (May 2022 to December 2023), the group consisted of 64 individuals, comprising 12 infants, 9 juveniles, 16 subadults, 20 adult females, and 7 adult males.

### 2.2. Behavioral Data Collection

Data were collected on 43 adult and subadult (age more than 3 years) langurs of both sexes for two consecutive years. Data collection was carried out over eight months in 2022 (May to December) and ten months in 2023 (January to December). The S Group was tracked from 06:00 to 18:00 by three observers (the authors HN and VM, and a field assistant) for a total of 326 days, averaging 20 days per month. Three times each observation hour (i.e., every 20 min), we conducted an instantaneous scan sample that could last up to 15 min [51], for a total of 3912 h of observation—240 h on average per month (2052 h in 2022 and 1860 h in 2023).

Throughout the 15 min scan, all state behaviors of the visible S Group members were recorded, noting the activity that individuals were engaged in when first spotted. Observations were conducted using binoculars from a distance of more than 10 m. Two observers (one author and the field assistant) were consistently present at the study site during observations. To ensure the reliability of our data, we assessed inter-observer agreement between the two observers, which yielded a high level of agreement (Cohen’s kappa = 0.70). Agreement values falling between 0.60 and 0.74 were considered indicative of good agreement [52] Given that observers were consistently present during all recorded behaviors, they were highly unlikely to have influenced the observed behavioral changes. 

The dogs were fully habituated to the presence of humans/researchers. Both dog and human presence were documented during each scan. If dogs and humans were within approximately 10 m of any S Group individual during a particular scan, they were noted as present; otherwise, they were considered absent. The selection of the 10 m distance for our study is the culmination of a decade-long research effort. We found that the S Group frequently inhabits areas where human activities, along with the presence of dogs, are common, such as villages, roads, and agricultural fields. At distances beyond 10 m, langurs tend to exhibit neutral behavioral responses in such settings. However, when humans and predatory dogs come into close proximity with langurs, their behavior is significantly affected. Therefore, we concentrated our investigation within this critical range to better understand these behavioral dynamics. Habitat-type codes were recorded for each scan sample, and further details of these codes are presented in the Appendix A. 

### 2.3. Behavioral Definitions

Using the comprehensive ethogram outlined by Dolhinow [53], we categorized langur behavior into four broad state behaviors, as detailed below:Feeding: When an animal is ingesting or masticating food. Locomotion: An animal moves, at any pace, from one location to another for a distance of more than 1 m. Resting: An animal remains stationary, hindquarters on the supporting surface, either asleep (i.e., eyes closed) or awake (i.e., alert, with eyes open). It may simultaneously engage in l vigilance behavior. Social: Social interactions like grooming, play, or sexual activity. 

### 2.4. Study Variables

*Presence of predatory dog(s)*: In all the models, a binary value of this variable is used to denote whether one or more dogs were present within a 10 m radius of the S Group or not. These free-roaming dogs may have belonged to villagers or been stray dogs within the S Group’s territory. *Presence of human(s)*: In all the models, a binary value of this variable is used to denote whether one or more humans were present within a 10 m radius of the S Group or not. Humans usually included villagers or occasional tourists within the S Group’s territory. *Habitat:* The study group’s habitat encompasses various areas, including dense forest patches, open patches near human settlements, and agricultural fields (Figure 1). Initially, the entire study area was categorized based on location and vegetation, which were later combined into two forest types—open forests and dense forests—across all models. Further details regarding these classifications are presented in Appendix A.

### 2.5. Model Formulation and Statistical Analysis

To investigate the potential impact of our study variables on the four major behaviors of CHLs, we conducted a Multinomial Logistic Regression Mixed Model (MLRMM) analysis using the nnet package [54,55] in R version 4.2.2 [56]. In MLRMM, the log odds provide an estimate of the relationship between predictor variables (habitat, presence of dogs, presence of humans) and the probabilities of each categorical outcome variable (the four types of behavior) in comparison to the reference category. Specifically, we used the log odds of feeding, locomotion, and social behavior relative to resting behavior as the baseline for all models. The selection of the reference category should be driven by theoretical or substantive considerations [57]. Thus, we use resting behavior as the reference category, given it is a dominant behavior in colobines for digesting their folivorous diet [58]. Additionally, selecting a reference category with a substantial number of observations can enhance the precision of estimates [59]. To control for the systematic variation in the presence and absence of humans and dogs that occurs over time, such as seasonal patterns or changes in human activity, the month was included as a random factor in all models. We conducted tests for outliers, the homogeneity of residuals, and variance inflation factors using the car package in R [60]. The results indicated no violations of the model assumptions. Specifically, all VIF values were below 3, suggesting no significant multicollinearity among the predictor variables [61]. Furthermore, we confirmed the validity and reliability of our modeling approach by using squared Mahalanobis distance residuals to diagnose various types of overdispersion [62], as well as conducting a score test to assess the goodness of fit of the most complex model [63].

MLR serves as an extension of binary logistic regression, accommodating more than two categories of dependent variables. MLRMM, like binary logistic regression, models the log odds of the outcomes as a linear combination of the predictor variables and employs maximum likelihood estimation to assess the probability of categorical membership, generating a unique set of parameters for each level of the dependent variable, referred to as regression coefficients (log odds) [64]. To understand how different factors affect the likelihood of each category of the outcome variable, we identified three predictor variables: presence of dog, presence of human, and habitat. We constructed nine potential models based on these variables and compared them based on Akaike’s information criteria (AIC). We applied information theory (IT) to identify the best model [65,66,67,68]. This method’s fundamental principle uses ΔAIC values to rank the model among the candidate set of models (Δi = AIC(i) − AIC (min)) [65]. Standardized model weights express the probability that a given model is the best among those in a set of models (Table 1). With the “aictab” function from the AICcmodavg package, we assessed the Akaike’s weight, or relative likelihood, and the cumulative weight of each model [69]. These metrics indicate the degree to which one model is more likely than another to explain variance in the data. AIC is the preferred method for model selection in ecology and evolution due to its effectiveness in addressing methodological challenges [70,71,72]. This method has been widely adopted in primate behavior studies across various species, including Japanese macaques [73] *Macaca fuscata fuscata*, vervet monkeys [74] (*Chlorocebus pygerythrus*), common chimpanzees [75]) *Pan troglodytes schweinfurthii*, black howler monkeys [76] *Alouatta pigra*, and mountain gorillas [77] *Gorilla beringei beringei*.

## 3. Results

### Selection of Activity Pattern

Among the nine models that were tested, the integrated model emerged as the most accurate in explaining the factors influencing the activity patterns of CHLs. This model demonstrated the lowest AIC value and the highest model probability, with a weight of 1 (as shown in Table 1). The integrated model incorporates all independent variables (dog presence, human presence, habitat type). In contrast, the remaining eight models, including the null model (as presented in Table 1), collectively exhibited a probability of less than 0.01.

In the presence of predatory dogs, the log odds value of adult CHLs engaging in feeding activities over resting activities was 0.006, whereas for locomotion and social activities, it was 0.202 and −0.455, respectively (see Table 2). Free ranging dogs significantly influenced the activity patterns of adult langurs, resulting in increased feeding and locomotion but decreased social activity when dogs were present in their habitat (see Figure 2a, Table 2; z = 1.698, z = 6.113, z = −11.375; *p* < 0.01).

Similarly, in the presence of humans, the log odds value of adult CHLs engaging in feeding activities over resting activities was 0.166, whereas for locomotion and social activities, it was 0.249 and −0.267, respectively (see Table 2). Adult langurs exhibited increased feeding and locomotion but decreased social activity when humans were present (Figure 2b, Table 2; z = 4.582, z = 7.875, z = −7.334; *p* < 0.01).

In open habitats, the log odds value of adult CHLs engaging in feeding activities over resting activities was 0.464, whereas for locomotion and social activities, it was 0.182 and −0.622, respectively (see Table 2). Habitat type significantly influenced the activity patterns of adult langurs, resulting in increased feeding and locomotion but decreased social activity in open habitats (Figure 2c, Table 2; z = 15.734, z = 6.261, z = −18.306; *p* < 0.01).

## 4. Discussion

### 4.1. Behavior Modifications in the Predator-Induced Landscape of Fear

Our results demonstrate that the presence of predatory dogs causes CHLs to alter their behavior, increasing their feeding and movement activity while lowering their social interaction. Colobine populations can differ in feeding and resting patterns, and these differences are more likely to be a result of behavioral adaptations than a byproduct of their shared evolutionary history [78]. By allocating maximum time to resting, colobines optimize their digestive efficiency and maximize the extraction of nutrients from their folivorous diet [58]. Under a landscape of fear, the presence of predators can shape the behavior and distribution of prey species even when predation itself does not occur [79,80,81]. Adjusting the allocation of foraging time is a common strategy for animals to mitigate the risk of predation [82]. Because animals respond to predators through complex, compensatory behavioral strategies, it is difficult to predict how these behaviors will ultimately impact fitness [83]. Prey species facing higher levels of predation risk may allocate more energy toward developing and maintaining anti-predator defenses or reducing other energetically expensive activities [84,85]. 

The natural Banj oak *Quercus leucotrichophora* forest, which is the preferred habitat for langurs [41], has become fragmented due to the high dependency of local people on the surrounding natural forest [86,87]. Langurs are increasingly dedicating more time to feeding on fodder crops in agricultural land, primarily due to the fragmentation of their preferred habitat [41,42,43]. Our results suggest that the differences in resource availability in modified agricultural landscapes relative to their preferred oak forests leads to greater feeding and moving, requiring them to expend more energy to forage and navigate their surroundings. These areas also do not provide many safe areas to rest and socialize. Based on 312 langur–dog interactions, all successful predatory attacks by dogs on S Group individuals occurred in agricultural fields, where langurs are notably vulnerable due to the lack of connected tree cover, providing a low chance of escape [18,43]. Our results suggest that despite the threat of predation, langurs increase their feeding time in predator-rich environments because these areas provide crucial resources for them. Similar behavior is also observed among herbivores in Africa; due to the distribution of grasslands and waterholes, grazers often have no choice but to utilize habitats located near the few available waterholes and grasslands, despite the high predation risk associated with these areas [88]. This behavioral adjustment reflects their prioritization of foraging and movement over social activities to maximize their fitness. Additionally, langurs employ high-cost anti-predator strategies such as giving alarm calls and engaging in fights with predators to mitigate the risk of predation [18]. Overall, our findings highlight the complex interactions between habitat fragmentation, foraging behavior, and predation risk in langurs, emphasizing their adaptive responses to changes in their environment to achieve optimal fitness.

### 4.2. Behavior Modifications in the Human-Induced Landscape of Fear

Similar to the presence of predatory dogs, the presence of humans caused CHLs to alter their behavior, increasing their feeding and movement activity while lowering their social interaction. This effect was partially spatial, as langurs were attempting to forage on crops and thus moving into agricultural areas where they were likely to encounter humans. Human presence can have significant effects on the behavior of various animals, for example, changing the foraging and vigilance behavior of elk *Cervus elephus* [89] and Nubian ibex *Capra nubiana* [90]. Human activities such as hunting, habitat modification, and disturbance can create perceived threats for wildlife, for example, in the case of red deer *Cervus elaphus* and wild boar *Sus scrofa*, human presence can influence their use of habitat and behavior [44]. The livelihood of local villagers in this region depends on agro-forestry crops [91] such as *Grewia optiva*, *Celtis australis*, and *Ficus virens*, which are also important parts of the CHL diet [49]. In response to the perceived threat posed by langurs to their crops and livelihoods, villagers engage in various negative interactions with the langurs, including throwing stones, using loud noises and firecrackers to chase them away, and even resorting to poisoning [41]. Human behavior and tolerance toward wildlife play crucial roles in shaping the landscape of fear in human-influenced environments [92]. In agroforest contexts in Africa, the economic value of cultivated food crops and the extent of damage caused by common chimpanzees *Pan troglodytes* can significantly influence human attitudes, tolerance, and behavior toward them [93]. Human tolerance toward CHLs can also influence the effectiveness of management and conservation efforts [41]. The negative attitude of the local community towards langurs is due to the langurs’ frequent consumption of their agricultural and fodder crops. To promote tolerance, it is critical to address the livelihood challenges faced by the local community. By integrating the livelihood needs of local communities into mitigation models, we can enhance their value perception of wildlife. This, in turn, facilitates community support for mitigating human–wildlife interactions through educational initiatives, habitat restoration, and the promotion of coexistence. Our findings contribute toward the understanding of how human behavior and tolerance affect the spatial distribution of CHLs and CHL behavior (i.e., the landscape of fear). This evidence can inform land-use planning and management strategies, such as the identification and establishment of corridors within the territory of the S Group. Consequently, we recommend integrating effective management strategies incorporating ecological and social dimensions of human–wildlife interactions. We can develop comprehensive and socially acceptable interventions by considering factors such as human behavior and tolerance alongside ecological variables like species ecology, population dynamics, and habitat preferences. 

### 4.3. Behavior Modifications in Habitats Associated with the Landscape of Fear

We found that CHLs exhibit an increase in feeding and locomotion activity and a decrease in social interaction and resting in open forest habitats. Primates in fragmented landscapes spend more time feeding to compensate for limited resources [94]. The home range of the S Group includes fragmented patches of dense forest, agricultural land, and human settlements, and they often move between the forest and agricultural fields for crop foraging [41]. Interactions with dogs typically occur in open forest patches or agricultural fields where continuous canopy cover is absent [18,42]. The fragmentation and loss of habitats is a major factor leading to the decline in many species, partly due to increases in predation by human-introduced predators [33]. Primates show changes in their activity budget, foraging and ranging in anthropogenically disturbed environments; for example, common chimpanzees *Pan troglodytes verus* prefer to rest, travel, and socialize in habitats with less human activity [35]. In anthropogenically disturbed areas, vervet monkeys spend significantly less time socializing and more time engaged in movement [95]. Similarly, toque macaques *Macaca sinica* in human-modified habitats allocated more time to moving and vigilance and less time to resting [96]. Based on the evidence presented here, we propose that the availability of food resources in open habitats plays a crucial role in shaping the behavior and ecology of CHLs, specifically by increasing feeding and locomotion and decreasing resting and social interactions. Future studies focusing on the relationship between food availability and primate behavior in agroforest landscapes have the potential to generate valuable insights that can contribute to conservation, management, and coexistence strategies. Identifying tree species and landscape features that sustain primate food sources while preserving crop yields can help agroforestry management practices establish a balance between agricultural productivity and biodiversity conservation efforts.

## 5. Conclusions

Our findings highlight that CHLs allocate more time to feeding and locomotion and less time to socializing in the presence of humans and predatory dogs. This behavioral shift is anticipated as an adaptive strategy aimed at increasing individual fitness in environments where predation risk and human disturbances are prevalent. These behaviors demonstrate their adaptive responses to the landscape of fear, where the mere presence of predators influences their behavior and resource utilization. Overall, our findings provide valuable insights into how CHLs adapt their behavior to deal with threats from humans and predators, ultimately contributing to our understanding of wildlife responses to anthropogenic pressures in shared habitats. In the long run, this adaptation may lead to a shift in CHL habitat use patterns, social dynamics, and foraging behaviors. Therefore, we propose the following practices. 1—Create safe corridors and protected areas where CHLs can access essential resources without facing significant threats from humans or dogs. 2—Conduct comprehensive research to gain deeper insights into CHL social structures, behaviors, and dynamics. 3—Engage local stakeholders in collaborative wildlife management and conflict resolution efforts. By implementing these measures, we can promote coexistence between humans and wildlife while mitigating the long-term impact of human disturbances and predation risk on wildlife populations. Overall, by establishing partnerships with communities, conservation organizations can promote sustainable practices, raise conservation awareness, and empower local communities to take ownership of conservation initiatives.

It is also essential to address the limitations of this study and propose opportunities for future research that could enhance our understanding and inform management practices more effectively. While the study reveals behavioral changes occurring within langurs in response to human and dog presence, it lacks direct evidence regarding the adaptability or long-term benefits of these changes for langurs. Therefore, further investigation comparing the fitness and survival outcomes of langur troops in various habitats (e.g., forest-dwelling vs. semi-urban) is crucial to assess the adaptive nature of these behavioral shifts. Such research could evaluate factors like reproductive success, survival rates, and overall population health to determine whether langurs in urban or semi-urban environments survive better despite the risks associated with human and dog interactions. These approaches would contribute significantly to strengthening management approaches and promoting coexistence.

## Figures and Tables

**Figure 1 biology-13-00410-f001:**
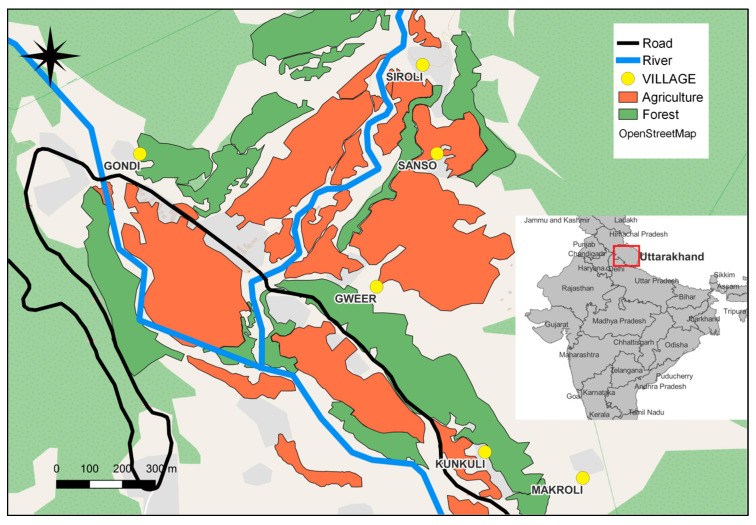
A study area map illustrating the spatial distribution of forested areas, villages, and agricultural land within the habitat of the S Group.

**Figure 2 biology-13-00410-f002:**
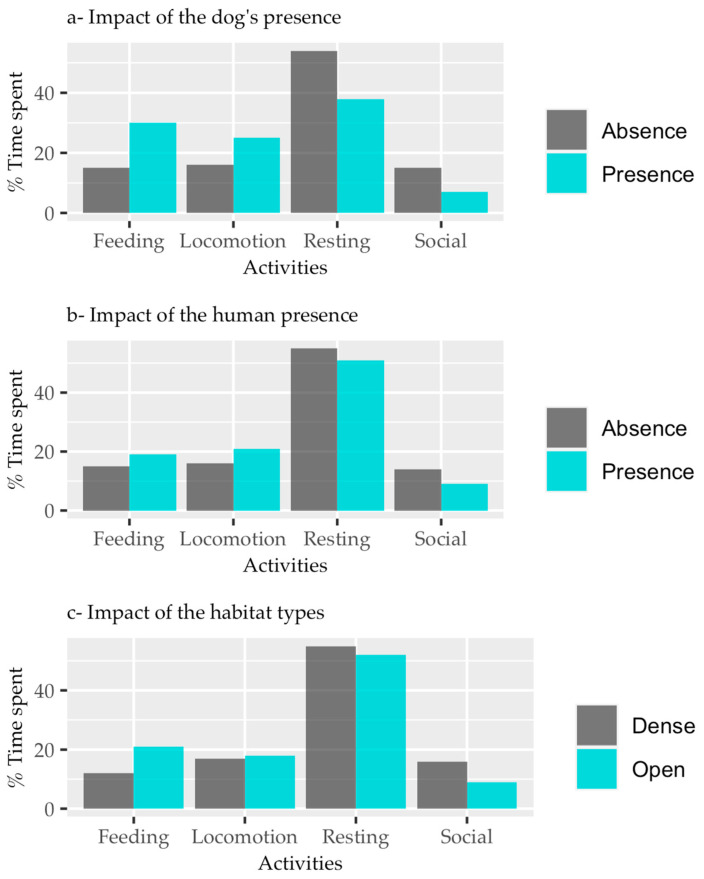
Predictor variables of (**a**) presence of dogs, (**b**) presence of humans, and (**c**) habitat type in relation to activity patterns.

**Table 1 biology-13-00410-t001:** Model selection based on AIC.

MLR Models	K	AIC	ΔAIC	Weight
PD + PH + H-OF + (1|Month)	24	154,160.8	0	1
PD + PH + PD*H-OF + PD*H-OF + H-OF (1|Month)	18	154,232.6	71.76	0
PH + H-OF + (1|Month)	9	154,564.3	403.52	0
PD + H-OF + (1|Month)	9	154,569.3	408.46	0
H-OF + (1|Month)	6	154,959.4	798.54	0
PD + PH + (1|Month)	9	155,579.5	1418.68	0
PD + (1|Month)	6	155,833.3	1672.46	0
PH + (1|Month)	6	155,888.2	1727.33	0
MLR~1 + (1|Month)	3	156,371.4	2210.53	0

AIC: The AIC value for each model; ΔAIC: difference in the AIC between the model with the lowest AIC and the following one; H-OF: habitat type—open forest; K: number of free parameters in the model; PD: presence of dog; PH: presence of human; Weight: model probabilities; *: interaction term.

**Table 2 biology-13-00410-t002:** Results from a multinomial logistic regression model.

Predictor Variable		Presence of Dogs	Presence of Humans	Habitat—Open Forest
Feeding	z value	0.601	11.255	25.641
	*p* value	<0.01	<0.01	<0.01
	*Coef (SE)	0.016 (0.027)	0.288 (0.026)	0.585 (0.023)
Locomotion	z value	6.135	8.122	4.717
	*p* value	<0.01	<0.01	<0.01
	Coef (SE)	0.156 (0.025)	0.202 (0.025)	0.106 (0.022)
Social	z value	−11.963	−5.407	−17.872
	*p* value	<0.01	<0.01	<0.01
	Coef (SE)	−0.386 (0.032)	−0.163 (0.030)	−0.492 (0.028)

*Coef (SE): Regression coefficient (standard error).

## Data Availability

All data needed to evaluate the conclusions in the paper are present in the paper and/or the Appendix A. The datasets analyzed during the current study are available from the corresponding author upon reasonable request.

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
