# Peer review of "Predatory Dogs as Drivers of Social Behavior Changes in the Central Himalayan Langur (Semnopithecus schistaceus) in Agro-Forest Landscapes"

_biology, 2024, doi:10.3390/biology13060410_

Round 1
Reviewer 1 Report
Comments and Suggestions for Authors
I had the pleasure of reviewing the manuscript titled: “Predatory Dogs as Drivers of Social Behavior Changes in the Central Himalayan Langur (Semnopithecus schistaceus) in Agro-forest Landscapes”. The authors investigated the impact of dogs, humans, and habitat on the behavior of Central Himalayan Langurs (CHL) in agroforest environments of Uttarakhand, India. They followed a habituated troop for several months and collected scan samples of the group’s behavior. They hypothesized that the presence of dogs and humans would affect the activity patterns of CHL, and that CHL would modify their behavior in habitats where these threats are more common. They tested their hypotheses by categorizing langur behavior into four broad categories: feeding, locomotion, resting, and social. They included three predictor variables: Presence of Dog, Presence of Human, and Habitat. They performed Akaike's information criteria (AIC) based model selection on 8 candidate multinomial regression models with behavior type as the dependent variable and different combinations of the aforementioned independent variables, including the full model and the null model. The results showed that the full model, which incorporates all independent variables (dog presence, human presence, and habitat type), was the best model explaining the factors influencing the behavior of CHL. In the full model, they found a similar pattern among the three independent variables: presence of predatory dogs, humans, and open forest habitat significantly influenced the activity patterns of adult langurs, resulting in increased feeding and locomotion but decreased social activity compared to resting. This study contributes to our understanding of how animals adapt their behavior to anthropogenic pressures to minimize risks while maximizing opportunities in shared landscapes. Such work can help develop more effective approaches for managing human-wildlife conflict and fostering sustainable coexistence.
The study is well presented, with proper grammar, adequate background information in the introduction section, and fairly comprehensive methods section. I have some questions about their method and analysis approach, and suggestions to improve it. I also think that the authors stretch the conclusions a bit to the point that may not be supported by their findings. Below are my comments in detail:
Methods and statistical analysis:
In behavioral data collection, they do not mention how far the researchers were to the troop, as their presence could also be a factor affecting CHL behavior. They include an approximate 10m distance as a condition for human presence, which may be reasonable, but I also wonder if it is too short. Please elaborate more on what steps were taken to ensure that researcher presence would not be a confounding factor. Also, they mention that three observers tracked the behavior, but it is not clear if all three were present at a time, or if only one or two of them were present on any given day. This again could be factor affecting CHL behavior depending on how far they were to the troop. Along these lines, a limitation of the methods is that they do not consider the number of dogs or humans but simply the presence of one or more dogs/humans. If the goal is to gain insights on effective management practices, a sense of how many dogs/humans are tolerable before affecting behavior to the point that it’s maladaptive would be necessary as it’s unlikely that one can propose a completely human/predator free solution that would be practical. Such insights are also necessary for tourism management recommendations. I believe it would not have costed much extra effort to note the number of dogs/humans. If they do have this data, I would recommend analyzing it with that information as that would provide insights not only on whether the presence affects behavior (which is fairly obvious that it will), but how much does it affect it.
A typo on line 157: “Three times…”
In the statistical analysis section, they do not mention that they had a random intercept for “Year” in their models, which is apparent in the Table 1. Given that the data collection included one half year and one full year (May 22- Dec 23), it is not clear to me that “Year” is a relevant category for the random effect. A more relevant confounding category would be the season or the month as those are more likely to affect CHL behavior than the year. I recommend reanalyzing after including month, or a relevant combination of months that represents local seasons as a random intercept e.g. May-Aug = summer, Sep-Nov = fall, Dec-Feb = winter etc. Another possibility is to consider the local planting season, as the reward associated with taking the risk to forage in the agricultural areas might depend on seasonal resource availability. I would let the authors decide the relevant random effect based on their knowledge of local conditions.
In Table 2, a major error is that the sign of the coefficients of social are wrong. Please make sure the table includes the correct signs of the coefficients.
Discussion and conclusions:
Overall, the authors do a good job of not over-interpreting the findings, but I suggest toning down a bit on management implications in one location. The conclusion that creating safe corridors or protected areas would help does not necessarily follow from the findings. The current findings only demonstrate that there is a change in behavior. They do not tell you anything about whether the change is adaptive. That would require a study that compares fitness of completely forest dwelling troops compared to semi-urban troops like the current study and see if one of them has greater fitness. It is possible that the urban troops have better fitness due to greater access to resources, even after accounting for predation and human conflict, and the risk of foraging with dogs/humans might be worth it. Hence, it’s possible that even if one were to create corridors, CHL would still prefer to take the risk in agricultural areas. It would be worth discussing the limitations of current findings and specific paths for future research that would be better equipped to inform management practices.
Reviewer 2 Report
Comments and Suggestions for Authors
This study offers some important insights into the impact of both dogs (and humans) on the behavior of the Nepal Gray langur and, given its significant conservation implications, has definite value and potential for publication in biology. However, in my opinion, the paper cannot be published in its current form. I stopped reading the manuscript at the end of the statistical analysis as several important pieces of information are missing. Furthermore, I suggest that the statistical analysis should be enhanced by adding interactions in the model to better understand the effect on the response variable. Specific comments are provided down below.
At the moment, I suggest major revisions. I would be happy to revise the new version of the manuscript.
Title
According to the official nomenclature listed on the IUCN Red List website, the common name of the species is ‘Nepal Gray langur’. Additionally, following the nomenclature rules established by the International Commission of Zoological Nomenclature, the scientific name of each species should be enclosed in parentheses only if the Genus of the species has changed over time.
These modifications should be applied throughout the entire text.
1. Introduction
Line 49. ‘(Canis familiaris)’. Please refer to the comment provided in the title.
Line 84. ‘(Otolemur garnettii lasiotis)’. Please refer to the comment provided in the title.
Line 85: ‘(Lemur catta)’. Please refer to the comment provided in the title. Furthermore, the ring–tailed Lemur is classified as ‘Endangered’ by the IUCN. It would be useful to specify.
Line 91. Please, change ‘have demonstrated’ with ‘have shown’. In this case, you are referring to monitoring activities conducted in the field. Scientifically, the term ‘demonstrated’ can be used only when referring to experiments conducted in a laboratory.
Line 98. ‘(Pan troglodytes verus)’. Please refer to the comment provided in the title. Moreover, change ‘demonstrated’ with ‘have shown’.
Line 107. ‘Central Himalayan Langurs’. Please refer to the comment provided in the title.
Line 123. What do you mean with ‘impact the activity of wild animals’? Changes in activity patterns? Habitat use? Spatio–temporal segregation? Please, specify.
Lines 124–128. The study was already conducted. Therefore, the sentence should be written in the past form.
Lines 128–132. Phrased in this manner (i.e., using the future tense), these are not hypotheses but predictions. For instance, a hypothesis could be: ‘The presence of dogs and humans induces a disturbance in the behaviour of Nepal Gray Langurs (NGLs)’. Accordingly, the prediction could be: ‘We expect to observe that NGLs will modify their activity patterns to reduce the likelihood of encounters with dogs and humans’. However, I suggest combining what you referred to as H1 and H2 into one hypothesis. There is no need to separate them.
2. Material and Methods
2.1. Study site and population
Lines 141–143. ‘The total home range of the S group encompasses both natural forests and agricultural fields, frequently bringing them into proximity with humans and predatory domestic dogs (Figure 1)’. From this sentence, it seems that some individuals were previously monitored using radio–tracking. Please cite the study conducted.
2.5. Model formulation and statistical analysis
This part is lacking several important information and some needs to be better explained and/or clarified.
1. Multinomial regression models rely on five main assumptions, i.e., linearity between the independent variables and the log–odds of the response variable, independence of errors (residuals), absence of multicollinearity among independent variables, large sample sizes (this is ok as the authors have many observations), and absence of outliers. Did the authors check the assumptions? If so, they should specify how. If not, the results may be unreliable.
2. Was the absence of overdispersion among data checked? How? Overdispersion may seriously affect the parameters estimates, including the p-value.
3. Was the goodness–of–fit of the most complex model tested? If so, how? This is another important aspect that should be addressed.
4. Based on the information provided in the text, it is unclear whether model selection was performed based on the AIC or AICc. However, the latter is typically used when sample sizes are low. This does not seem to apply to the authors, as they rely on a large dataset.
5. Because the authors introduced a random factor (i.e., Year), they are using multinomial logistic regression mixed models and not multinomial regression models.
6. It is unclear to me why the authors used only one habitat category. In the paragraph ‘2.4. Study variables’, they mentioned three main habitat categories comprising the areas, i.e., dense forests, fragmented forests (open forests), and agricultural lands. Why did the authors not include them as separate categories in the model? Moreover, given the large dataset, I believe that introducing some interactions to explore the potential effect of the presence of humans/dogs in relation to each habitat may provide useful information. An example could be as follows:
model <– response variable ~ PD:H-DF + PD:H-OF + PD:H-A + PH:H-DF + PH:H-OF + PH:H-A + PD + PH + H-DF + H-OF + H-A + (1 | Year)
Round 2
Reviewer 1 Report
Comments and Suggestions for Authors
I am happy with the author responses to my comments and recommend the manuscript for publication.
Author Response
Thank you for your positive feedback and recommendation for publication. We appreciate your time and effort in reviewing our manuscript. Your constructive comments have been valuable in helping us improve the manuscript.
Reviewer 2 Report
Comments and Suggestions for Authors
I appreciated the authors’ efforts to improve the manuscript. However, despite some clarifications being provided, I consider this effort still insufficient. The specific reasons are highlighted below:
· The scientific names of each species were not removed from brackets. As I mentioned in the previous version, the name should be placed in parentheses only when the Genus has been changed over time. The authors merely highlighted each scientific name in yellow without making the necessary changes.
· The authors did not provide an explanation for why they ranked the models based on the AICc rather than the AIC. They mentioned using the Information Theory Approach to rank the models. However, as I noted in the previous version, the AICc is used when the sample size is small (typically less than 30), which is not the case here.
· Including interactions among covariates/predictors can certainly affect the model's stability, complexity, and interpretability. Nevertheless, models are simplifications of reality, and in certain circumstances, including interactions is necessary to obtain a clearer understanding of the phenomenon under study. The large dataset at the authors' disposal can help increase model stability and improve parameter estimates. Furthermore, depending on the specific context, several optimizers (e.g., bobyqa, Nelder–Mead, etc.) can be used to enhance model convergence.
In conclusion, I strongly recommend that the authors attempt to include interactions and evaluate the results. Therefore, another round of major revisions is suggested.
Author Response
Reviewer(s)' Comments to Author and Author's Responses (AR)
AR: We are grateful to both Reviewers for their constructive comments and requests for clarification of some details. This has further helped us to improve the manuscript. Below, we respond in detail to each point made by reviewer 2. All changes to the text are highlighted in yellow and line numbers used to reference changes are provided below.
Reviewer 2
I appreciated the authors’ efforts to improve the manuscript. However, despite some clarifications being provided, I consider this effort still insufficient. The specific reasons are highlighted below:
1: The scientific names of each species were not removed from brackets. As I mentioned in the previous version, the name should be placed in parentheses only when the Genus has been changed over time. The authors merely highlighted each scientific name in yellow without making the necessary changes.
AR: Thank you very much for the constructive comment, we have corrected this in the revised manuscript.
2: The authors did not provide an explanation for why they ranked the models based on the AICc rather than the AIC. They mentioned using the Information Theory Approach to rank the models. However, as I noted in the previous version, the AICc is used when the sample size is small (typically less than 30), which is not the case here.
AR: Thank you very much for the constructive comment, we have reanalyzed the model selection based only on the AIC. However, the best model, according to the AIC is also the integrated model. Corrected values of AIC and ΔAIC are provided in Table 1 (Line 287).
3: Including interactions among covariates/predictors can certainly affect the model's stability, complexity, and interpretability. Nevertheless, models are simplifications of reality, and in certain circumstances, including interactions is necessary to obtain a clearer understanding of the phenomenon under study. The large dataset at the authors' disposal can help increase model stability and improve parameter estimates. Furthermore, depending on the specific context, several optimizers (e.g., bobyqa, Nelder–Mead, etc.) can be used to enhance model convergence.
In conclusion, I strongly recommend that the authors attempt to include interactions and evaluate the results. Therefore, another round of major revisions is suggested.
AR: Thank you very much for the constructive comment. We have included the interactions model in the AIC model selection. However, the interactions model is not ranked as the best compared to the other eight models. Among the nine models tested, the integrated model emerged as the most accurate in explaining the factors influencing the activity patterns of the Central Himalayan Langur. Revised results are presented in Table 1 (Line 287). Consequently, we performed the Multinomial Logistic Regression Mixed Model on the integrated model and presented the results and discussion in the revised manuscript.